# Cost Volume Meets Prompt: Enhancing MVS with Prompts for Autonomous Driving

## Abstract

Metric depth is foundational for perception, prediction, and planning in autonomous driving. Recent zero-shot metric depth foundation models still exhibit substantial distortions under large-scale ranges and diverse illumination. While multi-view stereo (MVS) offers geometric consistency, it fails in regions with weak parallax or textureless areas. On the other hand, directly using sparse LiDAR points as per-view prompts introduces noise and gaps due to occlusion, sparsity, and projection misalignment. To address these challenges, we introduce **Prompt-MVS**, a cross-view prompt-enhanced framework for metric depth estimation. Our key insight is to inject LiDAR-derived prompts into the cost volume construction process through a differentiable, matching-aware fusion module, enabling the model to leverage accurate metric cues while preserving dense geometric consistency provided by the MVS process. Furthermore, we propose depth-spatial alternating attention (DSAA), which combines spatial information with depth context, significantly improving multi-view geometric consistency. Experiments on KITTI, DDAD, and NYUv2 demonstrate the effectiveness of Prompt-MVS, which outperforms state-of-the-art methods by up to 34.6% in scale consistency. Notably, our method remains effective even with missing or highly sparse prompts and produces stable metric depth under severe occlusion, weak texture, and long-range scenes, demonstrating strong robustness and generalization. Our code will be publicly available.

## 1 Introduction

High-quality metric depth perception is critical for autonomous driving systems, enabling accurate obstacle detection, motion planning, and scene understanding. While cameras provide dense visual cues, monocular depth estimation suffers from inherent scale ambiguity, limiting its reliability in safety-critical applications. To recover metric-scale geometry, modern approaches have explored two promising directions: (i) leveraging sparse but accurate LiDAR or SfM points as geometric prompts to anchor absolute scale (Wang et al., 2025b; Lin et al., 2025; Guo et al., 2025); and (ii) exploiting multi-view stereo (MVS) frameworks that use calibrated camera sequences to enforce epipolar constraints and achieve scale-aware reconstruction (Cao et al., 2022; 2024; Izquierdo et al., 2025).

However, in real-world autonomous driving scenarios, both paradigms face fundamental limitations due to system-level constraints. First, LiDAR-based prompts are often spatially sparse and unevenly distributed—especially at long range or under adverse weather—leading to poor generalization when the prompt is missing or misaligned with the target region. Moreover, occlusions and sensor placement biases (e.g., forward-facing only) further degrade prompt coverage, resulting in unreliable scale anchoring regarding some blind areas. Second, while MVS offers a principled way to recover metric structure through multi-view cues, it critically depends on a sufficient baseline between views. In practice, ego-motion can be minimal during traffic jams, parking, or low-speed maneuvers—conditions where consecutive frames exhibit near-zero parallax. Under such degenerate configurations, epipolar geometry collapses, correspondences become ambiguous, and MVS performance degrades sharply, reverting effectively to monocular estimation with all its associated scale uncertainties.

This dilemma reveals a central tension: prompt-based methods offer *absolute scale* but lack *spatial completeness*, while MVS provides *dense multi-view consistency* but fails under *motion degeneracy*. Crucially, neither approach alone is robust across the full spectrum of operational conditions in

autonomous driving. Our core insight is that: metric depth should be recovered through a synergistic fusion of absolute scale anchors and relative multi-view cues, where each modality compensates for the other.

Building on this principle, we introduce **Prompt-MVS**, a unified framework that integrates sparse metric prompts with multi-view cues for robust depth estimation. Our approach leverages absolute scale from prompts while exploiting dense geometric consistency from multiple views. Technically, Prompt-MVS introduces three core components: (1) a confidence-aware multi-view aggregation scheme that modulates matching costs along the depth hypothesis axis using geometrically propagated prompts as soft priors, enabling reliable scale anchoring even with sparse inputs; (2) a prompt-count-agnostic fusion mechanism that dynamically weights views based on prompt availability and confidence, allowing well-informed views to regularize those without prompts through cross-view consistency; and (3) a depth–spatial alternating attention (DSAA) module that disentangles depth-level aggregation from 2D spatial modeling, preserving structural coherence while reducing computational overhead. Together, these designs enable Prompt-MVS to produce high-fidelity, 3D-consistent depth maps across a wide range of driving conditions. Experiments show state-of-the-art performance on challenging benchmarks, with significant improvements over both monocular and multi-view methods in depth accuracy and downstream 3D reconstruction quality.

Finally, our contributions can be summarized as follows:

- We propose **Prompt-MVS**, a unified framework that integrates sparse metric prompts with multi-view cost volume learning, achieving accurate depth estimation even in challenging scenarios in autonomous driving.

- We introduce a **confidence-aware multi-view aggregation** scheme, that dynamically modulates matching costs along the hypothesized depth axis as a soft depth prior, allowing views with reliable prompts to guide and compensate for less-informative views without prompts.

- We design a **depth–spatial alternating attention** (DSAA) module that alternately aggregates contextual information across depth hypotheses and spatial dimensions, keeping structural consistency in 2D space.

## 2 RELATED WORK

**Monocular Depth Estimation (MDE).** Monocular depth estimation (MDE) is a long-standing challenge in computer vision. Early approaches exploit image features and geometric heuristics (Hoiem et al., 2007; Saxena et al., 2008). With the development of deep learning, data-driven methods (Eigen et al., 2014; Guo et al., 2025; Fu et al., 2018) achieved significant performance gains, yet often suffer from poor generalization beyond their training domains. Recent advances have focused on improving open-world robustness. The DepthAnything family (Eigen et al., 2014; Yang et al., 2024) further leverages massive unlabeled imagery via semi-supervised training (Sun et al., 2024; Yang et al., 2023), producing strong zero-shot results across various real-world scenes. To further enhance generalization, several works scale up training data with diverse scene priors (Xian et al., 2018; 2020), adopt affine-invariant losses (Ranftl et al., 2020), or design stronger architectures based on vision transformers (Ranftl et al., 2021). Concurrently, the diffusion model has opened a new frontier for MDE. Marigold (Ke et al., 2024) adapts Stable Diffusion (Rombach et al., 2022) to predict depth from diffusion priors. DepthLab (Liu et al., 2024) introduces a "painting-style" generation strategy to sharpen structure. Depthfm (Gui et al., 2025) adopts flow-matching to accelerate sampling within diffusion frameworks. While these approaches recover compelling relative geometry, they typically inherit the scale ambiguity of generative models and thus fail to recover metric-consistent depth. To address this problem, classic methods supervise with RGB-D or LiDAR in specific domains (e.g., indoor scenes or street views) (Bhat et al., 2021; Yin et al., 2019), while more recent efforts (Butler et al., 2012; Guizilini et al., 2023; Kendall et al., 2018) demonstrate improved cross-domain generalization. ZoeDepth (Bhat et al., 2023) builds upon relative-depth pretraining (Birkl et al., 2023) and attaches domain-specific metric heads for scale adaptation. UniDepth (Piccinelli et al., 2024; 2025) jointly learns from metric and non-metric depth, further enhancing generalization. MoGe (Wang et al., 2025a) targets metric geometric estimation directly by exploiting the $z$-channel of predicted point clouds as a metric signal.

**Prior-based MDE.** In real-world scenarios (e.g., autonomous driving), cameras are often paired with LiDAR, motivating the use of sparse depth measurements as geometric priors for recovering dense, metrically calibrated depth maps (Schönberger et al., 2016; Jensen et al., 2014). Recent methods have explored various strategies to effectively fuse such priors. OMNI-DC (Zuo et al., 2024) addresses diverse sparsity patterns via a specialized architecture and probability-based loss, combined with scale normalization and mixed synthetic training. Marigold-DC (Viola et al., 2024) leverages generative models to capture complex visual statistics, improving cross-scene generalization. However, under extremely sparse priors, these methods can falter due to insufficient geometric guidance. To mitigate this, PromptDepthAnything (Lin et al., 2025) conditions a depth foundation model (Piccinelli et al., 2024) on a low-resolution depth prompt, while PriorDepthAnything (Wang et al., 2025b) injects explicit geometric constraints in a coarse-to-fine pipeline to retain performance even with scarce priors. DepthLab (Liu et al., 2024) first interpolates to fill holes and then refines with a depth-guided diffusion model, but effectiveness degrades when missing regions are large or the depth range is incomplete, as interpolation errors accumulate. Despite these advances, they remain vague and unconvincing. The core drawback is the heavy reliance on 3D-structured point cloud/depth prompting, which does not involve multi-view cues, thereby causing inconsistency.

**Multi-view Stereo (MVS).** Multi-view Stereo provides a complementary route to *metric*-aware depth: under calibrated cameras, depth can be triangulated by enforcing epipolar constraints. Early methods relied on patch-based matching for depth estimation (Schönberger et al., 2016; Furukawa et al., 2015). With the advent of deep learning, MVSNet (Yao et al., 2018) proposes an end-to-end formulation that decomposes MVS into feature extraction, cost-volume construction, and cost regularization, substantially improving accuracy and robustness. Subsequent works advance along several axes: architectural innovations for richer geometric and long-range context (Cao et al., 2022; 2024); explicit handling of occlusions and independently moving objects (Wimbauer et al., 2021; Long et al., 2020); efficiency-oriented designs that reduce compute and memory (Sayed et al., 2022; Yu & Gao, 2020); and joint estimation of camera poses and depth to mitigate extrinsic errors (Leroy et al., 2024; Wang et al., 2024). Murre (Guo et al., 2025) incorporates strong multi-view SfM priors into the diffusion process to enforce cross-view consistency. Despite these advances, most early methods are trained and evaluated within narrow domains, limiting their out-of-distribution generalization. MVSAnywhere (Izquierdo et al., 2025) addresses this by targeting cross-domain and cross-range robustness without requiring retraining on target data, which marks a shift toward foundation-model-style MVS systems. Nevertheless, MVS remains fundamentally constrained by epipolar geometry, which often fails in degenerate conditions (e.g., near-zero baseline, unchanged camera poses, or repetitive textures). In such cases, multi-view constraints provide little discriminative power and result in a collapse to an MDE problem with inherent scale ambiguity. The limitations of existing paradigms motivate a hybrid formulation that combines metric priors with MVS cues, balancing global scale awareness with local geometric consistency to improve robustness under degenerate configurations.

# 3 METHOD

**Overview.** The overall pipeline of Prompt-MVS is illustrated in Fig. 1. Given $N + 1$ calibrated images with known camera poses, we designate the reference image as $\boldsymbol{I}_0 \in \mathbb{R}^{H \times W \times 3}$ and its sparse metric depth prompt as $\boldsymbol{P}_0 \in \mathbb{R}^{H \times W}$, where $H \times W$ is the image resolution. The source views are $\{\boldsymbol{I}_i \in \mathbb{R}^{H \times W \times 3}\}_{i=1}^N$ with associated sparse depth prompts $\{\boldsymbol{P}_i \in \mathbb{R}^{H \times W}\}_{i=1}^N$. Conditioned on these inputs, Prompt-MVS predicts multi-scale depth maps for the reference view, progressively refining resolutions from $1/8$ to full scale $(1/1)$ of the original image size. Specifically, for the reference and source views $\{\boldsymbol{I}_i\}_{i=0}^N$, we adopt the first two blocks of a ResNet18 (He et al., 2016) as the feature encoder, mapping each image $\boldsymbol{I}_i$ to a deep feature map $\mathcal{F}_i \in \mathbb{R}^{\frac{H}{4} \times \frac{W}{4} \times C}$ at $1/4$ resolution, where $C$ is the channel dimension. The encoder weights are shared across views. Then, we discretize the depth range into $D$ hypotheses, $\{d_k\}_{k=1}^D$. For each source view $i$, the feature map $\mathcal{F}_i$ is warped into the reference camera at depth $d_k$. The warped features are concatenated with the reference feature $\mathcal{F}_0$ and auxiliary geometric metadata to form the initial cost volume $\mathcal{F}_{cv} \in \mathbb{R}^{\frac{H}{4} \times \frac{W}{4} \times D}$. To fully exploit prompt information across views, we first densify the sparse metric depth prompts $\{\boldsymbol{P}_i^d \in \mathbb{R}^{H \times W}\}$ and simultaneously estimate per-pixel confidence $\mathbf{Conf}_i^d \in \mathbb{R}^{H \times W}$. The resulting pseudo-dense prompts and their confidences are downsampled to $\frac{H}{4} \times \frac{W}{4}$ and used to refine the reliability of each spatial location in the cost volume, as depicted in Sec. 3.1. Moreover, to more effectively

Figure 1: **Overview of Prompt-MVS**. Given multi-view images, we first perform Semi-Densification to propagate sparse prompts and derive per-pixel confidence maps. (a) The source views are encoded into matching features, and a cost volume is constructed by correcting features across views over discretized depth hypotheses. (b) A key to performance is our confidence- and prompt- conditioned cost volume: the densified prompts and their confidences modulate and reweight the costs, suppressing spurious matches and sharpening responses along the depth axis. (c) Finally, the proposed Depth–Spatial Alternating Attention (DSAA) fuses monocular cues from the reference and source encoders with the spatial features extracted from the cost volume, explicitly disentangling depth and spatial dependencies for efficient and robust aggregation.

fuse mono-cues (from the reference image) with multi-cues (from cost volume), we introduce a Depth–Spatial Alternating Attention (DSAA) mechanism that will be depicted in Sec. 3.2.

## 3.1 SEMI-DENSIFICATION WITH CONFIDENCE

To make the projected sparse depth usable in image space, we first perform K-nearest-neighbor (KNN) interpolation for semi-densification. Let $D_s$ denote the sparse depth map with valid set, the image domain, and a sky mask from semantic segmentation. We completely skip sky and other semantics that do not require filling. In addition, to help the network distinguish original pixels from interpolated pixels and to better leverage the prompt information, we introduce an extra confidence map Conf whose weights are derived from each pixel's Euclidean distance to its nearest valid prompt. Furthermore, to account for the noise inherent in LiDAR, for each pixel $x$ where a sparse metric measurement $d_{obs}(x)$ is available, we assume it follows a Gaussian likelihood as follows:

$$\mathcal{N}(d(x); d_{obs}(x), \sigma_{obs}^2(x)), \tag{1}$$

where $\sigma_{obs}(x)$ encodes range-dependent sensor noise. Then we yield a closed-form posterior for $x$ through Bayesian fusion:

$$\sigma_{post}^{-2} = \sigma_{obs}^{-2} + \sigma_{prior}^{-2}, \quad \mu_{post}(x) = \sigma_{post}^2(x)\left(\frac{\mu_{prior}(x)}{\sigma_{prior}(x)} + \frac{d_{obs}(x)}{\sigma_{obs}^2(x)}\right). \tag{2}$$

For each pixel $x$ where a sparse metric measurement $d_m(x)$ is not available, we simply set $\mu_{post}(x) = \mu_{prior}(x), \sigma_{post}^2 = \sigma_{prior}^2$. We the define a numerically stable confidence as follows:

$$\text{Conf}(x) = \exp\left(-\frac{\sigma_{post}^2(x)}{\tau^2}\right), \tag{3}$$

where $\tau$ is temperature factor. Because the range of prompt depth values varies significantly across views and scenes, we first normalize the prompt depths. Specifically, we set the lower bound to 0.8 the original minimum and the upper bound to 1.2 the original maximum (to ensure the true depth range is covered). We use the same range to normalize the ground-truth depth maps. This

design ensures that the trained model predicts depths on the same scale as the input prompt, enabling effective use in subsequent multi-view feature extraction.

**Confidence-aware multi-view aggregation.** Following Fig. 1, we first build a feature volume and apply a matching MLP independently to each combined feature vector at a given spatial location and depth plane. This yields a raw cost volume $C_{raw} \in \mathbb{R}^{H/4 \times W/4 \times D}$ To leverage prompt maps and corresponding confidence from both the reference and source views, we further apply an aggregation MLP that fuses $C_raw$ with the dense prompts and confidence maps. Like the matching MLP, the aggregation MLP is shared across locations and depths and is evaluated in parallel at every plane of the volume. Concretely, for each voxel we form a 3D input vector

$$\mathbf{x} = [s, b_r, c_r, b_s, c_s], \tag{4}$$

where s is the matching score from $C_{raw}$, $b_r, b_s$ are the dense bias score of reference view and source view, defined as the absolute difference between the multi-view dense prompt depth and the current depth plane, and $c_r, c_s$ are the pixel-wise confidence. For pixels withou a valid prompt depth, we set $c = 0$ and $b = -1$ The aggregation MLP maps $\mathbf{x}$ to a fused score $\tilde{s} = \mathrm{MLP}_{agg}(\mathbf{x})$. This procedure is used identically at training and test time.

## 3.2 DEPTH–SPATIAL ALTERNATING ATTENTION

In multi-view stereo (MVS), the cost volume exhibits pronounced anisotropy along the depth and spatial dimensions: along the depth dimension $d \in \{1, \ldots, D\}$ it enumerates multiple geometric hypotheses for the same pixel and evaluates their consistency, whereas along the spatial dimensions $(u, v) \in \{1, \ldots, H\} \times \{1, \ldots, W\}$ it encodes cross-pixel structural priors and occlusion cues, such as edge continuity and occlusion boundaries at discontinuities. Flattening a 3D voxel volume into a single token sequence and applying global self-attention is computationally prohibitive and, more critically, conflates two fundamentally different dependencies: depth-wise correlations across hypothesized planes and in-plane spatial structure. Inspired by VGGT's Alternating Attention (AA), which factorizes video attention into intra-frame followed by inter-frame, we redefine the attention scope to alternate across orthogonal axes, applying depth-first attention and then spatial attention in a stacked manner. This axis-wise, divide-and-conquer factorization reduces the complexity of global attention and prevents dependency entanglement, thereby enabling more effective fusion of geometric priors along the depth axis with spatial context within each plane and yielding more stable optimization. As illustrated in Fig. 1, within the reference-image encoder we convert the cost volume into tokens using two cross-row convolutional layers. Before tokenization, the cost-volume features are concatenated with monocular features taken from the first two encoder stages (at $1/4$ and $1/8$ of the input resolution), after appropriate transposition and linear projection. The module outputs a sequence of $H/16 \times W/16$ tokens, matching the sequence length of the monocular stream. We then feed these tokens into a ViT-B initialized with DINOv2 weight. To fuse geometric evidence from the cost volume with appearance cues from the image, we adopt an alternating attention scheme. After a linear projection that maps both streams to a shared embedding space, the cost-volume tokens and the reference-image tokens first undergo intra-stream self-attention independently; their results are then summed (residual fusion) and subsequently processed by inter-stream self-attention to exchange information across streams. We repeat this procedure at the 2nd, 5th, 9th, and 11th transformer blocks, thereby integrating monocular cues at multiple depths of the ViT. This design enables effective coupling of depth (along the hypothesis dimension) and spatial information, improving the joint reasoning over geometry and appearance.

## 3.3 IMPLEMENTATION DETAILS

**The Design of Metadata.** Following MVSAnywhere (Izquierdo et al., 2025), we construct a metadata vector per pixel–depth hypothesis and feed it to a source–view–agnostic MLP to construct the cost volume. Specifically, each metadata token contains seven components: the feature dot product between warped source and reference features, ray directions, reference-plane depth, reprojected depths, relative ray angles, relative pose distance, and a depth validity mask. This design enables fixed-dimensional tokenization that is independent of the number of source views, facilitating scalable and flexible cost aggregation across inputs with varying view counts.

**The selection of keyframes.** For reference–source selection, we adopt the strategy proposed by MVSAnywhere for dense video sequences. To ensure robustness in sparse frame settings, we further

enhance the selection criterion by forming reference–source tuples based on geometric overlap. Specifically, we select non-consecutive frames that maximize view intersection—measured by reprojection coverage and visibility—while maintaining sufficient parallax. This balances completeness of multi-view correspondence with depth discriminability, thereby improving depth estimation reliability under large temporal gaps.

**Loss function.** We adopt the supervision losses from MVSAnywhere (Izquierdo et al., 2025). Specifically, we apply an $\ell_1$ loss between the logarithm of the ground-truth depth and the logarithm of the prediction, together with gradient and surface-normal losses. The training objective is imposed at four decoder output scales.

$$\mathcal{L}_{depth} = \frac{1}{HW} \sum_{s=1}^{4} \sum_{i,j} \frac{1}{s^2} \left| \uparrow_{gt} \log D_{pred}^{i,j} - \log D_{gt}^{i,j} \right|,$$

$$\mathcal{L}_{grad} = \frac{1}{HW} \sum_{s=1}^{4} \sum_{i,j} \left| \nabla \downarrow_s \frac{1}{D_{pred}^{i,j}} - \nabla \downarrow_s \frac{1}{D_{gt}^{i,j}} \right|, \tag{5}$$

$$\mathcal{L}_{normal} = \frac{1}{2HW} \sum_{i,j} \left( 1 - N_{pred}^{i,j} \cdot N_g t^{i,j} \right),$$

$$\mathcal{L}_{loss} = \mathcal{L}_{depth} + \mathcal{L}_{grad} + \mathcal{L}_{normal},$$

where $D_{pred}$ is the network output depth map, $D_{gt}$ is the ground truth depth map, $\nabla$ denotes first-order spatial gradients, $N_{pred}$ is the normal map derived from predicted depth and camera intrinsics, $N_{gt}$ is the normal map derived from ground truth depth and camera intrinsics, and $i, j$ superscripts indicate pixel indices. Here $\uparrow_{gt}$ denotes that each output is aligned with the full-size ground truth depth map via nearest-neighbor upsampling and $\downarrow_s$ represents downsampling to scale $s$.

## 4 EXPERIMENTS

### 4.1 EXPERIMENTAL SETUP

**Datasets.** We evaluate Prompt-MVS in a non–zero-shot (in-dataset) setting on two real-world autonomous-driving datasets, training and validating on each dataset separately. The evaluation targets outdoor driving scenes with diverse image resolutions, sparse-depth densities, sensing configurations, and noise characteristics. KITTI (Geiger et al., 2012) comprises driving sequences with paired RGB images and sparse LiDAR depths at a resolution of $1216 \times 352$. Semi-dense ground truth is obtained by temporally accumulating multiple consecutive LiDAR sweeps with error filtering. We adopt the official validation split with 1000 samples and remove outliers in the guidance points by comparing each point to the local minimum depth within a $7 \times 7$ window. DDAD (Guizilini et al., 2020) features a $360°$ multi-camera rig and long-range LiDAR with depth measurements up to 250m The official validation set provides 3950 samples per camera at a resolution of $1936 \times 1216$. In our experiments, we use only the front-facing camera and randomly subsample 20 of the available depth measurements as sparse input, applying the same LiDAR filtering procedure as in KITTI.

**Evaluation Protocol.** Following recent work (Viola et al., 2024), we report four commonly used metrics to compared $\hat{d}_i$ and GT depth $d_i$. The Mean Absolute Error(MAE), Root Mean Squared Error(RMSE), Absolute Relative depth (AbsRel), and inlier percentage $\tau$ are defined as follows:

$$\text{MAE} = \frac{1}{N} \sum_{i \in \Omega} |\hat{d}_i - d_i|, \quad \text{RMSE} = \sqrt{\frac{1}{N} \sum_{i \in \Omega} (\hat{d}_i - d_i)^2},$$

$$\text{AbsRel} = \frac{1}{N} \sum_{i \in \Omega} |\hat{d}_i - d_i|/d_i, \quad \tau = [\max(\hat{d}/d, d/\hat{d}) < 1.05], \tag{6}$$

where $[\cdot]$ is the Iverson bracket.

**Training details.** We train our model on four A100 GPUs for 70K steps, with an effective batch size of 12 (6 samples per GPU, accumulated over two forward passes), at input resolution $640 \times 480$. Gradient accumulation is applied to stabilize optimization under limited hardware resources. We

Table 1: Benchmark study of depth estimation methods. All methods are evaluated on *KITTI* (Geiger et al., 2012) and *DDAD* (Guizilini et al., 2020), respectively, and each data set contains two metrics. All the metric are reported in meters. The **best** and second best scores are highlighted in **bold** and underline.

| Method | Venue | KITTI | | DDAD | | Avg | |
|---|---|---|---|---|---|---|---|
| | | MAE↓ | RMSE↓ | MAE↓ | RMSE↓ | MAE↓ | RMSE↓ |
| NLSPN (Park et al., 2020) | ECCV'20 | 1.335 | 2.076 | 2.498 | 9.231 | 1.917 | 5.654 |
| CompletionFormer (Zhang et al., 2023) | CVPR'23 | 0.952 | **1.935** | 2.518 | 9.471 | 1.735 | 5.703 |
| DepthLab (Liu et al., 2024) | arXiv'24 | 0.921 | 2.171 | 4.498 | 8.379 | 5.425 | 5.275 |
| Marigold (Ke et al., 2024) | CVPR'24 | 1.765 | 3.361 | 22.872 | 32.661 | 12.319 | 18.011 |
| PromptDA (Lin et al., 2025) | CVPR'25 | 0.934 | 2.803 | 2.107 | 7.494 | 1.521 | 5.149 |
| PriorDA (Wang et al., 2025b) | arXiV'25 | 1.705 | 4.083 | 4.745 | 12.330 | 3.225 | 8.207 |
| MVSAnywhere (Izquierdo et al., 2025) | CVPR'25 | 1.704 | 3.562 | 4.182 | 11.539 | 2.943 | 7.551 |
| **Ours-A** (only sparse prompt) | - | 0.769 | 2.626 | 2.211 | 6.114 | 1.490 | 4.370 |
| **Ours-B** (only semi-densify prompt) | - | 0.802 | 2.451 | 2.426 | 6.776 | 1.614 | 4.614 |
| **Ours-C** (only DSAA) | - | 1.102 | 2.318 | 3.126 | 8.036 | 2.114 | 5.177 |
| Ours | - | **0.602** | 2.218 | **2.024** | **5.976** | **1.313** | **4.097** |

adopt component-wise learning rates: the matching encoder and cost-volume MLP use an initial learning rate of $2e^{-4}$ until step 70k, then $2e^{-5}$ thereafter. The depth decoder and the mono/multi cue combiner are initialized at $1e^{-4}$ and linearly decay to $1e^{-7}$. The reference image encoder starts at $1e^{-5}$ and linearly decays to $1e^{-8}$. A weight decay of $2e^{-4}$ is applied across all parameters. We initialize the model from MVSAnywhere (Izquierdo et al., 2025) pretrained weights. For the first 10k steps, we adopt a stage-wise schedule: all parameters are frozen except MLP layers and the DSAA module, allowing rapid adaptation of prompt-conditioned components without destabilizing the backbone. Afterward, we unfreeze all modules and perform end-to-end fine-tuning. To improve robustness and generalization to missing or noisy prompts, we regularize the prompts during training with stochastic dropout and additive noise. Concretely, we randomly drop prompt regions with a fixed probability and inject zero-mean noise into both the prompt values and their confidence maps, discouraging over-reliance on any single prompt and encouraging cross-view consistency.

### 4.2 EXPERIMENTAL RESULTS

**Evaluation on KITTI and DDAD Dataset.** We first validate the proposed method on KITTI and DDAD datasets. During test stage, images are resized to $640 \times 480$ and the number of source views is fixed to 7. We use OneFormer (Jain et al., 2023) to obtain a sky mask, which is applied during inference to suppress unreliable matches in sky regions. Evaluation follows the official protocol, and we report MAE and RMSE over valid pixels. As summarized in Tab. 1, Prompt-MVS delivers substantial gains over the MVSAnywhere baseline without prompts in KITTI dataset, reducing MAE by 64.67% and RMSE by 37.73% overall. As for DDAD dataset, Prompt-MVS achieves an average MAE of 2.024 and RMSE of 5.976, outperforming all competing methods. Moreover, compared with prompt-based monocular depth estimators, our multi-view formulation leverages geometric constraints and cross-view consistency to achieve lower overall error than both classical and learning-based monocular baselines. Qualitative results in Fig. 2 further show that Prompt-MVS produces more accurate depth with markedly fewer outliers.

**Evaluation on Static Scenes.** To assess robustness in such motion degenerate regimes, we evaluate Prompt-MVS on KITTI 2011_09_0026 and DDAD 000193 sequences. As reported in Tab. 2, we compare methods using MAE, RMSE AbsRel, and the inlier rate $\tau$. Prompt-MVS surpasses the baseline method MVSA (Izquierdo et al., 2025) in KITTI scene and has the largest inliers. While in DDAD scene, Prompt-MVS outperforms all competing methods across every metric, indicating that our multi-view prompt fusion is better suited to real-world autonomous-driving scenarios and remains robust under motion degeneracy.

### 4.3 ABLATION STUDY

**Effects of Proposed Components.** To verify the effectiveness of Prompt-MVS, we conduct ablation studies on key components of our pipeline. As shown in Tab. 3, we evaluate the model under various configurations. The comparison between Ours-A and Ours-B highlights the importance

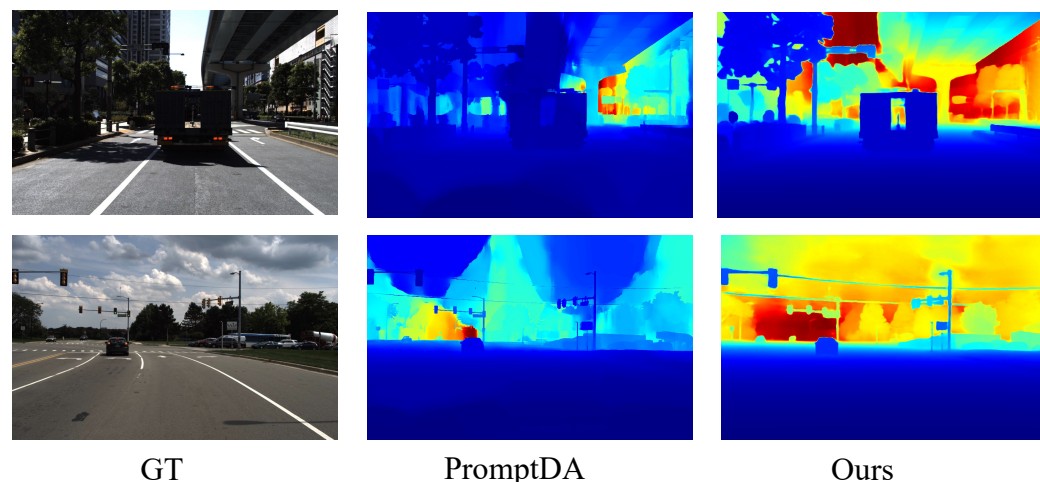

|  GT | PromptDA | Ours |

Figure 2: Experiment results on DDAD (top) and KITTI (bottom). Our method has better visual results compared to LiDAR-prompt monocular depth estimation methods.

Table 2: Benchmark study of depth estimation methods on static scenes. All methods are evaluated on 2011_09_29_0026 sequence of *KITTI* (Geiger et al., 2012), and 000193 sequence of *DDAD* (Guizilini et al., 2020), respectively, and each data set contains four metrics *MAE* and *RMSE* are reported in meters, *AbsRel* and $\tau$ are reported in percentage. The **best** and second best scores are highlighted in **bold** and underline.

| Method | KITTI-Static | | | | DDAD-Static | | | |
|---|---|---|---|---|---|---|---|---|
| | MAE↓ | RMSE↓ | AbsRel↓ | $\tau$↑ | MAE↓ | RMSE↓ | AbsRel↓ | $\tau$↑ |
| Depth Pro (Bochkovskii et al., 2024) | 0.971 | **2.122** | 0.064 | 60.029 | 6.851 | 14.677 | 0.206 | 9.131 |
| PromptDA (Lin et al., 2025) | **0.872** | 2.365 | **0.045** | 72.666 | 7.676 | 17.573 | 0.209 | 22.085 |
| PriorDA (Wang et al., 2025b) | 1.324 | 3.395 | 0.084 | 67.009 | 5.961 | 13.536 | 0.226 | 42.224 |
| MVSAnywhere (Izquierdo et al., 2025) | 8.890 | 11.231 | 0.655 | 15.132 | 5.146 | 13.229 | 0.160 | 47.973 |
| Ours | 3.439 | 10.289 | 0.096 | **81.468** | **2.585** | **8.867** | **0.068** | **84.208** |

of prompt densification—a simple yet critical step that enriches sparse metric priors and improves their spatial coverage. Furthermore, the results from Ours-B and Ours-C demonstrate that both the confidence-aware aggregation and the DSAA modules contribute meaningfully to performance. When all components are combined, the full model achieves the best results, confirming the synergistic design of Prompt-MVS.

**Effects of Prompt Absence.** To simulate common LiDAR blind spots in autonomous driving scenarios, we evaluate depth prediction performance when prompts are absent. Specifically, we remove the point cloud prompts from the reference view and discard prompts from 4 out of the 7 source views. As shown in Fig. 3, compared to the baseline, our method maintains strong performance in this degraded setting, with only marginal degradation in depth accuracy relative to the full-prompt case. This demonstrates the robustness of Prompt-MVS to missing geometric priors.

**Effects of Prompt Density.** We further conduct experiments on the impact of the prompt density. We sample the LiDAR prompt of DDAD dataset with different configurations, i.e., 64 lines and 16 lines. As shown in Tab. 3, since denser prompts provide more accurate metric prior, our method consistently outperforms the baseline approach and achieves significant results under the guidance of a higher number of LiDAR lines, revealing the upper limit of our approach.

## 5 CONCLUSION

This paper presents **Prompt-MVS**, a unified framework for robust metric depth estimation that effectively bridges the gap between sparse geometric prompts and multi-view consistency. By syner-

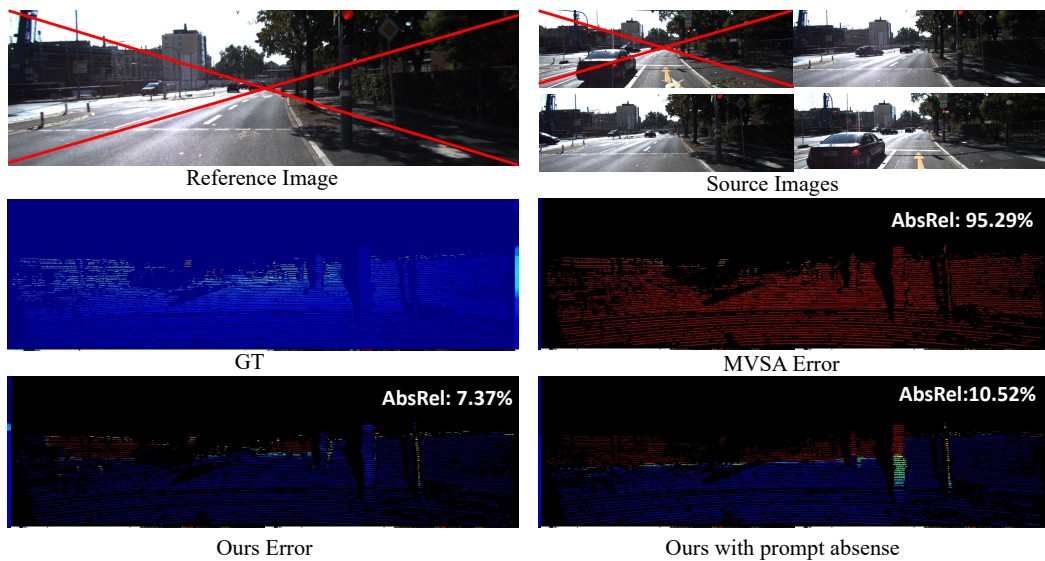

Figure 3: Prompt Absence Cases. We discard the prompt of reference image and part of the source view images. Our method surpass baseline method (Izquierdo et al., 2025) with limited degradation comapred to the full-prompt case. Red cross refers to ignore prompt.

Table 3: Ablation study of Prompt Density. All methods are evaluated on the validation sequence of *DDAD* (Guizilini et al., 2020). Each data set contains four metrics *MAE* and *RMSE* are reported in meters, *AbsRel* and $\tau$ are reported in percentage. The **best** scores are highlighted in **bold**.

| Method | DDAD-64 Lines | | | | DDAD-16 Lines | | | |
|---|---|---|---|---|---|---|---|---|
| | MAE↓ | RMSE↓ | AbsRel↓ | $\tau$↑ | MAE↓ | RMSE↓ | AbsRel↓ | $\tau$↑ |
| PromptDA (Lin et al., 2025) | 2.669 | 8.672 | 0.628 | 78.924 | 3.432 | 9.800 | 0.082 | 70.106 |
| PriorDA (Wang et al., 2025b) | 2.653 | 8.268 | 0.055 | 80.583 | 3.407 | 9.532 | 0.078 | 66.551 |
| **Ours** | **1.583** | **5.405** | **0.049** | **87.488** | **1.870** | **5.862** | **0.061** | **81.935** |

gistically fusing absolute scale priors with dense epipolar constraints, our approach overcomes the fundamental limitations of each modality: it remains reliable under motion degeneracy where MVS fails, and generalizes well even with sparse or incomplete prompting. We design a confidence-aware fusion mechanism and a depth–spatial alternating attention module which leverage the complementary strengths of absolute scale anchoring and multi-view cues, achieving superior performance under challenge scenarios in autonomous driving such as motion degeneracy and partial observability. Experiments show state-of-the-art results in depth accuracy and 3D reconstruction quality, demonstrating the effectiveness of structured cross-modal integration for real-world 3D perception.

**Limitations.** Our method assumes calibrated cameras and accurate pose estimates, which may not always be available in wild environments. Additionally, the performance may degraded considering temporal coherence and consistency, due to the mono-depth decoder. Future work may explore the spatial-temporal consistency and extension to non-calibrated scenes.

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
