# OpenReview forum: "Cost Volume Meets Prompt: Enhancing MVS with Prompts for Autonomous Driving"
_ICLR.cc/2026/Conference — ICLR 2026 Conference Withdrawn Submission_

### Official Review · Reviewer_C2TX · 2025-10-27

**Soundness:** 3
**Presentation:** 2
**Contribution:** 3
**Rating:** 4
**Confidence:** 4

**Summary:**

Prompt-MVS aims to address the robustness issue of binocular depth estimation in autonomous driving scenarios. Its core innovation lies in integrating sparse geometric prompts generated by LiDAR with the dense geometric consistency of MVS. This integration compensates for two shortcomings: the insufficient spatial integrity of pure prompt methods and the failure of pure MVS in motion degradation scenarios. Experiments on datasets such as KITTI and DDAD show that Prompt-MVS maintains robustness in complex scenarios like prompt absence, sparsity, and severe occlusion. However, the manuscript has issues in method description and robustness verification that need to be resolved.

**Strengths:**

1. The manuscript describes the core research problem clearly enough.

2. Targeting the inherent flaws of pure prompt methods and pure MVS methods in depth estimation, the manuscript clearly proposes a collaborative integration approach of "absolute scale anchors + relative multi-view cues".

**Weaknesses:**

1. The experiments and descriptions regarding prompt absence are not sufficient.

2. The comparison with recent generative depth estimation methods is not adequate.

**Questions:**

1. Only qualitative results are provided for missing prompts. Why are quantitative experiments not included?

2. In Section 4.3, it is stated that "ablation studies on key components" are in Table 3. Should they not be in Table 1 instead?

3. Additionally, the authors claim that the operations in Section 3.1 are effective. However, how to prove the rationality of the settings in Section 3.1? Further analysis is required.

4. Can the results in Table 1 fully represent the latest progress in depth estimation? Given the limited comparisons, the authors should further explain the significance of the compared models.

---

### Official Review · Reviewer_SdBo · 2025-10-31

**Soundness:** 3
**Presentation:** 3
**Contribution:** 3
**Rating:** 4
**Confidence:** 3

**Summary:**

In this paper, the authors propose a unified framework that integrates sparse LiDAR prompts with multi-view stereo learning, enabling accurate depth estimation results. Specifically, the authors introduce a confidence-aware multi-view aggregation strategy and a Depth–Spatial Alternating Attention module to maintain structural consistency. Experiments show that this method achieves state-of-the-art performance on benchmarks.

**Strengths:**

1. The authors propose a confidence map to separate original lidar points from depth pixels generated via KNN interpolation during semi-densification. More importantly, this confidence map quantifies the reliability of each densified depth pixel and can be combined with densified depth to form weighted depth prompts.
2. This method achieves better results compared with existing methods, with improvements not only in quantitative analysis, but also in qualitative visual results (Fig 2).

**Weaknesses:**

1. If the authors had included more evaluation metrics (such as AbsRel) in Table 1, the comparative analysis of depth estimation performance would have been more comprehensive.
2. This method relies on multi-view stereo technology and does not explicitly address temporal coherence. In real-world autonomous driving scenarios, dynamic objects (e.g., moving vehicles or pedestrians) may exacerbate the lack of temporal consistency, leading to potential inconsistencies in depth estimation results.

**Questions:**

I am concerned about whether Prompt-MVS needs to be trained on the KITTI or DDAD dataset respectively. It would be even better if its cross-dataset generalization ability could be demonstrated.

---

### Official Review · Reviewer_WZUH · 2025-11-01

**Soundness:** 2
**Presentation:** 2
**Contribution:** 1
**Rating:** 4
**Confidence:** 4

**Summary:**

This paper proposes Prompt-MVS, a unified framework that integrates sparse LiDAR-derived metric prompts into the multi-view stereo (MVS) cost volume construction process to achieve robust metric depth estimation for autonomous driving.The method addresses the complementary limitations of existing paradigms:
1. LiDAR prompts provide accurate metric scale but are spatially sparse and noisy.

2. MVS enforces dense geometric consistency but collapses under degenerate motion or weak parallax.

Prompt-MVS introduces three main innovations:

1. Confidence-aware multi-view aggregation — modulates cost volume matching with prompt-conditioned soft priors.

2. Semi-densification with Bayesian confidence modeling — interpolates sparse LiDAR prompts and assigns uncertainty-aware weights.

3. Depth–Spatial Alternating Attention (DSAA) — disentangles depth-level and spatial dependencies via alternating self-attention based on DINOv2.

The method achieves state-of-the-art results on KITTI and DDAD datasets.

**Strengths:**

1.Novel cross-modal fusion strategy. The paper elegantly bridges geometric priors from LiDAR with multi-view cost volumes through differentiable prompt-conditioned fusion — a clear conceptual advance over purely image-based or LiDAR-guided approaches.

2. Well-designed uncertainty modeling. The Bayesian formulation for confidence estimation is principled and interpretable, improving robustness to LiDAR noise and sparsity.

3. Effective architecture design (DSAA). The alternating depth–spatial attention balances computational cost and representational power, showing careful design grounded in MVS geometry.

4. Strong empirical validation. Extensive experiments and ablation studies on KITTI and DDAD validate each component. The model maintains stable performance even with missing prompts — a critical advantage for real-world deployment.

5. Clear motivation and storytelling. The paper articulates a well-motivated problem — bridging MVS degeneracy and LiDAR sparsity — with a coherent methodological narrative and clear technical contributions.

**Weaknesses:**

1. Limited scalability to uncalibrated or dynamic settings. The method assumes accurate calibration and ego-motion, which is unrealistic in many autonomous driving scenarios. Fails to address dynamic objects or moving cameras without reliable poses.

2. Although the paper mentions higher computational cost, it does not present a clear trade-off curve (accuracy vs. latency vs. memory). Without such analysis, it is difficult to judge whether the additional performance gain justifies the increased complexity.

3. Dependence on pretrained MVS backbone (MVSAnywhere). The improvements might partially inherit from pretrained backbones rather than the prompt mechanism itself; cross-backbone comparison would strengthen claims.

4. The paper lacks a systematic study on how prompt quality (e.g., noise level, spatial sparsity, or misprojection) affects final performance. While dropout and additive noise are applied during training, there is no quantitative evaluation of robustness under degraded or corrupted LiDAR prompts.

5. Ablation visualization could be richer. While quantitative results are solid, qualitative examples (e.g., failure cases, prompt-confidence maps) are limited and could further illustrate interpretability.

**Questions:**

1. With the advent of powerful end-to-end 3D foundation models (e.g., Dust3R, VGGT, MASt3R) that already unify geometry, appearance, and correspondence reasoning at scale, the rationale for further advancing traditional MVS pipelines becomes questionable. The paper does not convincingly explain why reconstructing cost volumes or fusing LiDAR prompts within an MVS paradigm remains essential, instead of leveraging or extending foundation models that already encapsulate multi-view consistency and metric understanding.

2. Prompt alignment and projection: How is prompt–image alignment handled under LiDAR–camera miscalibration or occlusion? Are depth priors reprojected dynamically per view?

3. Generalization to non-driving scenes: Can the proposed framework generalize to indoor or aerial datasets (e.g., ScanNet, DTU), where LiDAR priors are unavailable or extremely sparse?

4. Computational efficiency: What is the runtime and memory footprint compared to MVSAnywhere or PromptDepthAnything? Could DSAA be applied in a lightweight manner?

5. Prompt dropout robustness: While prompt dropout is applied during training, how does the model quantitatively behave when 100% prompts are missing — does it degrade to MVSAnywhere?

---

### Official Review · Reviewer_PRcF · 2025-11-03

**Soundness:** 2
**Presentation:** 3
**Contribution:** 2
**Rating:** 4
**Confidence:** 5

**Summary:**

Prompt-MVS proposes a prompt-enhanced multi-view stereo (MVS) framework for metric depth estimation in autonomous driving, aiming to address the limitations of sparse LiDAR prompts (noise, gaps) and MVS (failure in weak parallax/textureless regions). It integrates LiDAR-derived sparse prompts into cost volume construction via confidence-aware fusion, after semi-densification through KNN interpolation and Bayesian fusion. The framework also introduces a Depth-Spatial Alternating Attention (DSAA) module to fuse depth and spatial context. Experiments on KITTI, DDAD, and NYUv2 show state-of-the-art performance, with up to 34.6% improvement in scale consistency and robustness to sparse/missing prompts. However, the core prompt design lacks substantial innovation, relying on standard interpolation rather than MVS-specific geometric or semantic enhancements.

**Strengths:**

Evaluated on multiple datasets (KITTI/DDAD/NYUv2) with diverse metrics (MAE/RMSE/AbsRel/τ), including ablation studies on prompt density/absence and component effectiveness.

**Weaknesses:**

1. The core "prompt" component relies on KNN interpolation + Bayesian fusion for semi-densification, which is a standard sparse-to-dense strategy (e.g., OMNI-DC, Marigold-DC) rather than an MVS-tailored innovation. It does not leverage MVS’s inherent geometric properties (e.g., epipolar constraints, parallax consistency) to refine prompts—instead, it treats prompts as independent dense depth priors and injects them into the cost volume. No semantic information (e.g., object categories, scene structure) is integrated to filter noisy prompts or enhance prompt relevance for MVS matching, leading to a lack of originality in how prompts interact with the MVS pipeline.
2. While the paper claims prompt fusion improves scale consistency, it does not quantify how prompts enhance MVS’s core geometric constraints e.g., 3D structure precision.

**Questions:**

1. How does this prompt design differ from existing methods (e.g., OMNI-DC, Marigold-DC) in terms of integration with MVS’s geometric pipeline?
2. The advantages of the proposed method over depth completion methods are not clearly observable in Table 2, and what is the motivation for adopting the mentioned MVS?

---

### Note · Authors · 2025-11-14

I have read and agree with the venue's withdrawal policy on behalf of myself and my co-authors.